

# Toward a clinical real time tissue ablation technology: combining electroporation and electrolysis (E2)

Enric Guenther[1,2,3,*], Nina Klein[1,3,4,*], Paul Mikus[1], Florin Botea[5,6], Mihail Pautov[5,6], Franco Lugnani[7], Matteo Macchioro[7], Irinel Popescu[5,6], Michael K. Stehling[1,2,3] and Boris Rubinsky[1,2]

[1] Biophysics, Inter Science GmbH, Gisikon, Lucerne, Switzerland
[2] Department of Bioengineering and Department of Mechanical Engineering, University of California, Berkeley, Berkeley, CA, USA
[3] Institut fur Bildgebende Diagnostik, Offenbach, Germany
[4] Department of Information and Communication Technologies, Universitat Pompeu Fabra, Barcelona, Spain
[5] Center of General Surgery and Liver Transplantation, Fundeni Clinical Institute, Bucharest, Romania
[6] Center of Translational Medicine, Fundeni Clinical Institute, Bucharest, Romania
[7] Hippocrates D.O.O. Divaĉa, Malaga, Spain
* These authors contributed equally to this work.

Corresponding author
Enric Guenther,
enric@interscience.ch

## ABSTRACT

**Background:** Percutaneous image-guided tissue ablation (IGA) plays a growing role in the clinical management of solid malignancies. Electroporation is used for IGA in several modalities: irreversible electroporation (IRE), and reversible electroporation with chemotoxic drugs, called electrochemotherapy (ECT).
It was shown that the combination of electrolysis and electroporation—E2—affords tissue ablation with greater efficiency, that is, lower voltages, lower energy and shorter procedure times than IRE and without the need for chemotoxic additives as in ECT.

**Methods:** A new E2 waveform was designed that delivers optimal doses of electroporation and electrolysis in a single waveform. A series of experiments were performed in the liver of pigs to evaluate E2 in the context of clinical applications. The goal was to find initial parameter boundaries in terms of electrical field, pulse duration and charge as well as tissue behavior to enable real time tissue ablation of clinically relevant volumes.

**Results:** Histological results show that a single several hundred millisecond long E2 waveform can ablate large volume of tissue at relatively low voltages while preserving the integrity of large blood vessels and lumen structures in the ablation zone without the use of chemotoxic drugs or paralyzing drugs during anesthesia. This could translate clinically into much shorter treatment times and ease of use compared to other techniques that are currently applied.

## INTRODUCTION

Image guided tissue ablation (IGA) plays an important role in the oncological management of primary (e.g., hepatocellular carcinoma, HCC) and secondary liver malignancies (e.g., metastases), both rank among the most frequent types of cancer worldwide (*Ananthakrishnan, Gogineni & Saeian, 2006*). Whilst surgical resection remains the standard of care, guidelines recommend IGA, for the treatment of early stage HCC due to good outcomes (*Bruix & Sherman, 2011*), resulting in a growing interest for these techniques for medical applications (*Galle et al., 2012*).

Two modalities can be considered well established by now: radiofrequency (*Gazelle et al., 2000*) and microwave ablation (*Yu & Liang, 2017*). Less frequently used are laser-induced thermal therapy (*Stafford et al., 2010*), cryoablation (*Gage, Baust & Baust, 2009*) and focused ultrasound (*Kennedy, 2005*). All of them are thermal ablation methods (*Deng, Hong & Stegemann, 2016*). Whilst relatively well understood and fairly easy to handle, their main disadvantages are relatively long ablation times (min), limited ablation volumes and unreliable tumor cell death near larger vessels due to the heat sink effect (*Lu et al., 2002*; *Yu et al., 2008*; *Wright et al., 2005*). Due to heat diffusion, radiofrequency and cryoablation have a transition zone where tissue gets damaged but not all (cancer)-cells reliably killed (*Bhardwaj et al., 2009*). The elevated temperature which thermal ablation technologies apply, affect every molecule in the treated zone. Therefore, they not only destroy malignant tumors but also normal anatomical structures such as nerves, viscera and most vessels, requiring prudent use in areas of the body with vital anatomical structures (*Bhardwaj et al., 2009*).

Recently, a novel, non-thermal tissue ablation method has been introduced, irreversible electroporation (IRE) (*Davalos, Mir & Rubinsky, 2005*). IRE employs ultra-short (microsecond) strong electric field pulses (up to 2500 V/cm) to induce cell death via irreversible permeabilization of the cell membrane (*Rubinsky, Onik & Mikus, 2007*). Due to the selective interaction of the electrical fields with the cell membranes only, in which they lead to pore formation, non-cellular tissue elements such as fibers, basal membranes and interstitial matrix, are not affected by IRE and thus preserved; a unique property of IRE (*Rubinsky, Onik & Mikus, 2007*; *Edd et al., 2006*). This endows IRE with low toxicity compared to thermal ablation methods (*Charpentier et al., 2010, 2011*). It has been shown that nerves, vessels and viscera, albeit denuded of cells by IRE, are structurally preserved and remain functionally intact by re-population of their fibrous scaffold by ingrowth of cells from outside the ablation field (*Phillips, Maor & Rubinsky, 2010, 2011*; *Phillips et al., 2012*).

The parameters typically used for clinical IRE are summarized in *Martin et al. (2016)*. IRE is delivered between needle electrodes that brace the targeted lesion. The minimum number of pulses given for each electrode pair is typically 70 and the typical range is between 90 and 100 (*Martin et al., 2016*). The pulses are given at a frequency of 1 Hz, with an interval of several seconds between each decade of pulses. The applied electric fields range from 1,400 to 2,000 V/cm with a maximum of 3,000 V/cm. The ablated zone is often elliptic, with a maximal distance between electrodes of 2.0 cm and a maximal electrode

exposure length of about 1.5 cm. The relatively small ablation zones requires the use of several electrode pairs and meticulous placement of several electrodes in specified geometries. For larger tumors, treatment with IRE may require the re-positioning of the electrodes to produce several merging ablation fields to cover the whole lesion. While each electroporation pulse is microseconds long, both electrode re-positioning and the application of 90 electrical pulses between each electrode pair renders the ablation process lengthy, in the order of magnitude of tens of minutes.

Despite IRE's limitations in terms of lesion size and procedure length, it has shown promising results for the treatment of malignancies of the liver (*Kourounis et al., 2017*; *Narayanan et al., 2013*; *Thomson, Kavnoudias & Neal, 2015*), pancreas (*Martin et al., 2016*; *Martin, 2013*; *Martin et al., 2015*) and prostate cancer (*Fritz et al., 2017*; *Murray et al., 2016*; *Narayanan, 2015*). IRE selectively affects only the cell membrane, while the extracellular matrix in the treated region remains intact (*Rubinsky, Onik & Mikus, 2007*; *Phillips et al., 2012*). IRE has gained prominence on the strength of its ability to treat tumors in cases where traditional minimally invasive ablation and surgical resection are unavailable. IRE is particularly valuable in treatment of patients with tumors adjacent to vital structures such as nerves (*Onik, Mikus & Rubinsky, 2007*), blood vessels (*Maor et al., 2010*), major biliary and hepatic structures (*Kourounis et al., 2017*), the rectum (*Srimathveeravalli et al., 2013*) and bladder (*Li et al., 2017*).

Recently, the group of Marshall et al. reported that the current delivered during IRE and ECT electroporation protocols also generates products of electrolytic reactions in addition to electroporation (*Maglietti et al., 2013*). This finding has led to recommendations to design protocols that reduce the extent of electrolysis during electroporation to avoid the effects of electrolysis (*Maglietti et al., 2013*; *Turjanski et al., 2011*; *Olaiz et al., 2014*). However, products of electrolysis are actually useful for ablation and are used as a method for tissue ablation by themselves (*Nilsson et al., 2000*). While effective, electrolytic tissue ablation protocols are lengthly and can last for hours. Nevertheless, tissue ablation by electrolytic ablation, also known as Electrochemical Therapy, shares some of the unique attributes of IRE; it can be used near large blood vessels in the liver and pancreas without harming the lumen (*Gravante et al., 2011*). This has led to the suggestion that rather than eliminating electrolysis from electroporation protocols designed to ablate tissue, it may be actually beneficial to combine electrolysis and electroporation to enhance tissue ablation (*Phillips et al., 2015a*, *2015b*).

It was shown that delivering electrical energy in a form that produces a judicious combination of electroporation and electrolytic effects can be more effective at tissue ablation than either electroporation or electrolysis alone (*Phillips et al., 2015a*, *2015b*; *Stehling et al., 2016*). It was also shown in both small (*Phillips et al., 2016*) and large animal (*Stehling et al., 2016*; *Klein et al., 2017*; *Rubinsky et al., 2016*) experiments that it is possible to design an effective minimally invasive tissue ablation protocol that employs a single waveform, which delivers simultaneously both electrolysis and electroporation (E2) (*Phillips et al., 2015b*; *Klein et al., 2017*). The combination of electrolysis and reversible electroporation, named E2, can be employed for low energy, large volume tissue ablation. E2 takes advantage of the synergistic effects of electrolytically produced cytotoxic chemical

compounds, their driven diffusion along an electric field gradient and their facilitated entry into cells by electroporation of cell membranes.

In consideration of the demands for an ideal tissue ablation modality, namely effectiveness, speed, large ablation volumes and low toxicity, and in view of the published fundamental work towards reducing the process of electroporation and electrolytic production and distribution into one wave form, we hypothesized that it should be feasible to generate a single E2 waveform that can meet these clinical demands with a generator prototype designed for that purpose. This study on pig liver was designed to verify the performance of the generator and the clinical relevant outcome of the E2 waveform ablation.

To better assess the technical feasibility of an E2 waveform, a series of goals of clinical relevance in a pig liver model was formulated. The goals were:

1. Evaluation of the performance of the E2 waveform generator;
2. Find minimum electrical charge per volume required for complete tissue ablation between the electrodes (bridged ablation);
3. Determine minimum number of electrical waveforms required for complete tissue ablation between the electrodes (bridged ablation);
4. Determine the effect of the electric field magnitude on the ablation protocol;
5. The effect of E2 waveforms on important tissue structure (blood vessels, bile ducts)
6. Understanding additional mechanisms of damage superimposed on the damage due to the combination electroporation and electrolysis;
7. Temporal histology after treatment with an E2 waveform;
8. A single simulation of a typical clinical procedure.

# MATERIALS AND METHODS

## Animal model

Experiments were conducted in compliance with the ethical and legal framework imposed by national legislation of Romania and the European Union. The experimental protocol was approved by the Ethics Committee of Fundeni Clinical Institute as well as by the Bucharest Sanitary-Veterinary Authority (no. 316). Subjects of the study were three in vivo 80 kg female pigs. After being fasted for 24 h, each animal was pre-medicated with a combination of acepromazine (0.5 mg/kg) and ketamine (15 mg/kg) via intramuscular injection. Anesthesia was intravenously administered in form of Propofol (2.5 mg/kg) and 0.1 mg Fentanyl. After endotracheal intubation anesthesia was maintained with sevoflurane in 80% oxygen (adjusted to 2–2.5% End-tidal sevoflurane). Postoperative pain was treated with morphine 0.1 mg/kg IM and ketoprofen 1 mg/kg after 6 h. Cefazolin 25 mg/kg IV was administrated every 2 h. The experiments were performed on open liver through a surgical window with sterile electrodes. Each electrode placement was attended using conventional photo cameras and ultrasound imaging (Hitachi Prius, Surgical Transducer, Steinhausen, Switzerland) as shown in Fig. 1. In this study, a total of 19 lesions

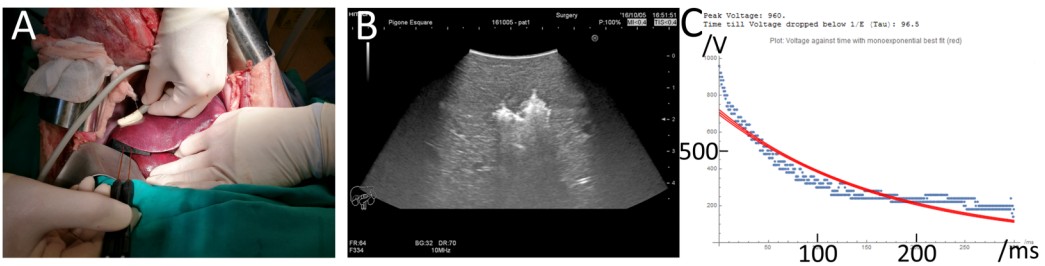

**Figure 1 Experimental protocol.** (A) Insertion of two electrode needles in the liver. (B) Ultrasound image of the treated tissue immediately after the delivery of the E2 waveform. The bright white area in the middle is due to ultrasound waves reflected by electrolytic gas produced from the E2 waveform and approximately congruent with final ablation zone as seen on pathology 24 h after. (C) Actual voltage vs. time signature of the E2 waveform (blue) vs. a theoretical single exponential decay waveform (red) that delivers the same charge as the E2 waveform.

were induced in the three pig livers. The parameters as well as pathology images of the 19 lesions can be found in the https://doi.org/10.6084/m9.figshare.8061827.v1. The surgical window was closed after completion of the test protocols and the animals were kept alive for 24 h, 72 h and 2 h with the pain prevention protocol described above. No muscle relaxation was used through the entire experiments.

## Generator

There is no commercially available device or system which is designed to produce waveforms in the required shape, amplitude and safety. Hence, a prototype was designed to carry out the experiments. The top level electrical circuit is shown in Fig. 2. The generator is designed to quickly charge selectable capacitor banks, decouple them and release the charge with selectable parallel and serial resistors. All switches are controlled by a custom microcontroller board which again is run by a computer interface. The charging of the capacitors is realized with a 3 kV quick charging power supply, the discharge anode and cathode can be set to any of 10 output ports with externally attachable applicators (electrodes). Before the application of the treatment waveform, the software was configured to deliver a pre-pulse which was set to 500 V for these experiments. This served the purpose of measuring the tissue impedance but also pre-distribute electrolysis. The measured impedance allows for adjustment of voltage and capacitance; however, this automatic adaptation was deactivated for the experiments presented here. The electrodes used were proprietary design needle-type titanium alloy electrodes with a diameter of 1 mm, with an adjustable exposure length of up to 5 cm. A 3D-printed spacer block (5 cm width, 1 cm length, 1 cm depth) was used which had a diameter of 1.2 mm and holes every 0.25 mm to allow parallel spacing of the electrodes in 0.25 mm distances. The electrodes were inserted into the according holes at equal depth to have 1 cm of shielded overstand at the bottom additional to the exposed length (usually 1.5 cm). The electrodes were bent 90° on top of the spacer block for stability purposes during the experiments. The bottom of the spacer block was pressed gently against the liver lobe after it was ensured that the exposed part of the electrodes (Inter

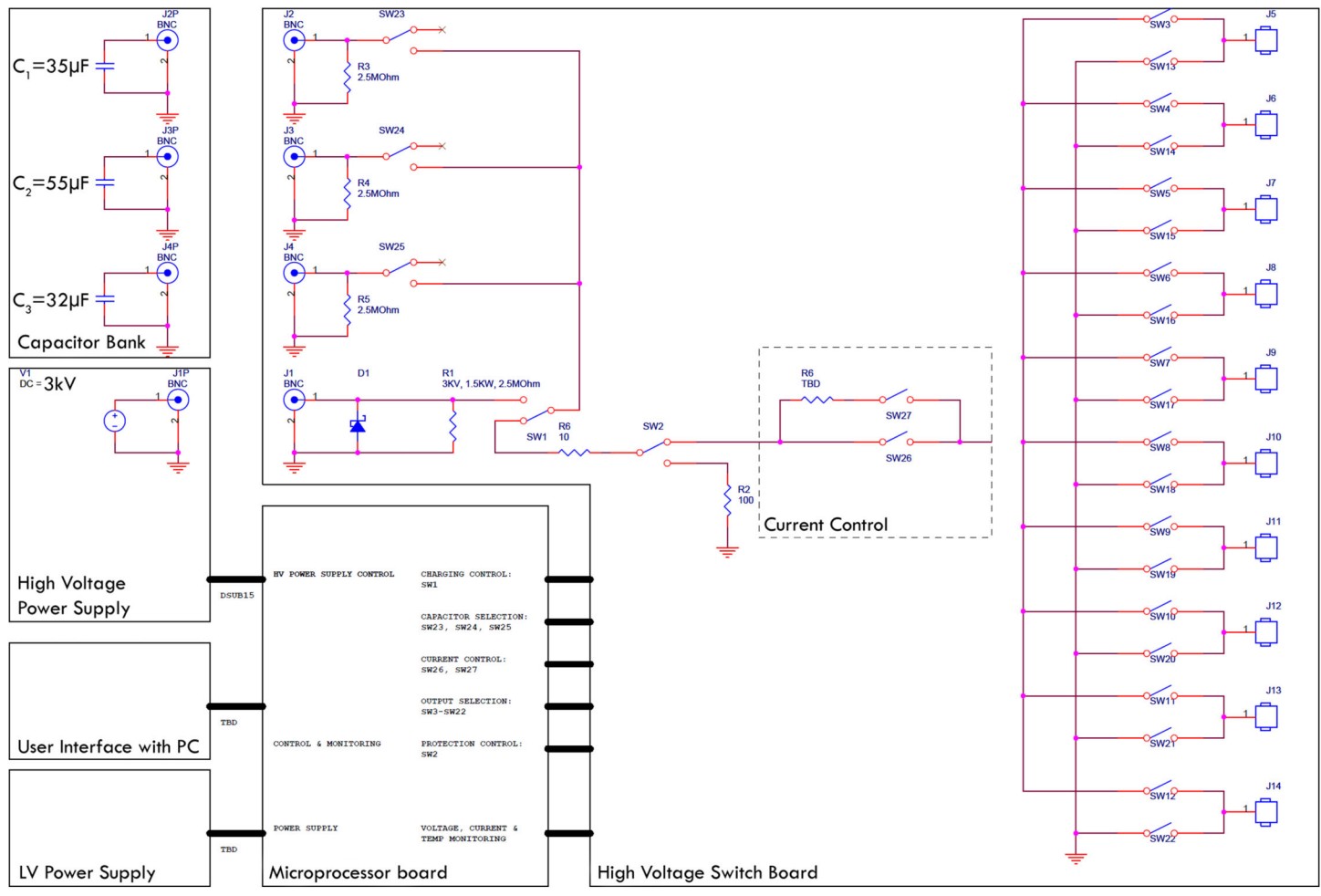

**Figure 2 Top level block diagram of the E2 generator electrical design employed.**

Science GmbH) was entirely inside liver tissue. Both electrodes and spacer are shown in Fig. 1A.

## Measurements

Every experiment was monitored and recorded using ultrasound (Hitachi Prius with EUP-334 Intraoperative Fingertip Convex Transducer; Hitachi Ltd., Tokyo, Japan) on video and/or photo. Each waveform was measured by clamping a high voltage differential probe set to 1/100 voltage reduction (Probe Maser 4241A 70MHZ, 7000V-5000VRMS; Probe Maser Inc., El Cajon, CA, USA) between ground and the conductor at the output of SW26 (See Fig. 1). The differential probe was attached to an oscilloscope (Rigol DS1054 50MHz 1GSa/s; RIGOL Technologies Inc., Beaverton, OR, USA) in single shot mode set to output a csv file. The csv file was then exported via USB to a PC running a custom script based on Wolfram Mathematica 10.0 (Wolfragm Reseach, Hanborough, UK) for visualization, tau time approximation, current calculation and storage as shown in Fig. 1C. From the best-fit models, one can calculate the current, as the capacitance is defined by the
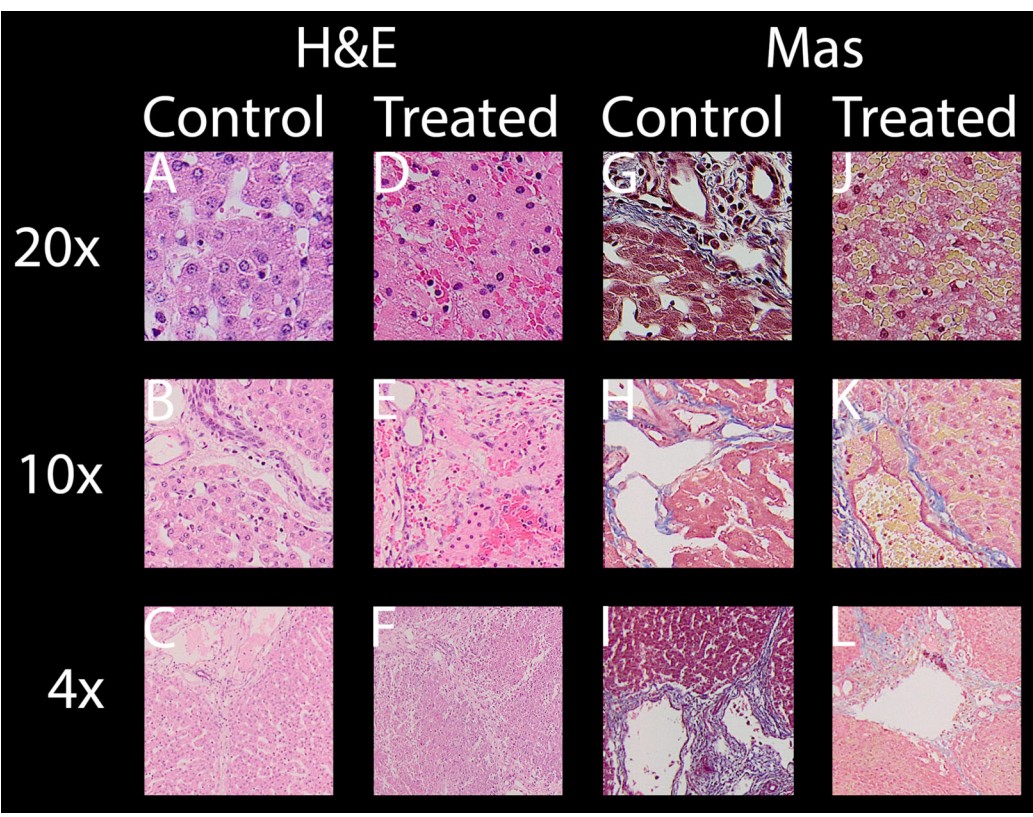

**Figure 3** For both stainings employed, H & E (A–F) and Mas (G–L), where each has a column showing healthy "control" micrographs (A–C and G–I) and "treated" (D–F and J–L) tissue micrographs with the magnification in the accordingly rows labeled 20× (A, D, G, J), 10× (B, E, H, K) and 4× (C, F, I, L).

capacitor selection before the E2W I = C * dV/dt where I is the current in Ampere, C the capacity in Farad and dV/dt the first time derivative of the voltage fit model.

## Histological examination

After liver treatment, the pigs were kept alive for 2, 24 and 72 h before euthanization with KCl 7.45% 1 ml/kg. The livers were harvested and flushed with saline solution as previously described (*Rubinsky et al., 2016*), the lesions were bread loafed perpendicularly to the liver surface and soaked in 10% phosphate buffered formalin for 10 h until being embedded in paraffin blocks. For microscopical analysis, 3 μm sections were cut from the samples and stained with Hematoxylin & Eosin (H&E) or Masson's trichrome (Mas) staining. Micrographs were taken with an Olympus optical microscope with xc50 5 Megapixel cooled color camera.

All evaluated images are available in the https://doi.org/10.6084/m9.figshare.8061827.v1. In the following paragraphs, ablation will be evaluated by looking at distinctive features of H&E and Mas as well as macroscopic impression of the lesion. Figure 3 shows selected typical excerpts from pathology scans at different magnifications to demonstrate the distinctive changes after 24 h compared to a non-effected control. The pathologically distinctive features indicating ablation in non-thermal electroporation-based treatments

are more subtle, because cell death is induced and not caused by melting and carbonization of volumes of tissue. Most prominent changes frequently mentioned and also statistically summarized in Table 1 are:

**Condensed or pyknotic nuclei**—Most clearly visible on 20 and 10× magnification on H&E stained example slides of Fig. 3 by comparing the size and density of the nuclei (dark-purple circles) but also at least structurally visible on all other magnifications and staining.

**Condensation of hepatocyte cytoplasm**—Area around the nuclei appear significantly denser. Most clearly visible on H&E staining where the nuclei are easily identifiable by their dark-purple color.

**Hemorrhage**—Identifiable by dense pink areas on the H&E slides and yellow areas on the Mas slides visible on all magnifications.

**Edema** – Identifiable by dilated sinusoidal spaces best visible on 4× magnifications as white fragments or cracked structures.

**Damaged sinusoids**—Identifiable by the absence of the regularly rounded "white" areas on the 20x magnified H&E and Mas example images.

**Cellular structure disruption**—Cells, most easily identifiable by their nuclei in the images, in healthy tissue are usually found in systematic arrangements to each other, forming tissue structures. In some control images at 20×, even segmentations (cell membrane) might be visible, see for example H&E control image at 20× magnification of Fig. 3. In treated tissue samples, one can witness the loss of spaces and structural arrangements of the cells to each other in all magnifications and stainings.

**Vessel and/or bile duct integrity**—Whilst the dynamics of tissue response to E2 is likely to be at least as complicated as with IRE (*Rubinsky, Onik & Mikus, 2007*) and its characterization was not a goal of this study, intact collagen structures of vessels (especially larger vessels) and ducts in the treatment field is worth noting. Tissue selectivity is a major advantage of electroporation-based treatments, allowing it to be used near major vessels other than thermal based ablation energies. Superficial vessels and duct integrity can be observed both in H&E and Mas staining while the connective tissue is best visible by employing Mas staining.

## RESULTS

### Evaluation of the performance of the E2 waveform generator

Figure 1 provides abbreviated information on the experimental protocols of this study. Details are found in the materials and methods section. In a typical experiment, the E2 waveform was delivered between two electrodes inserted in the liver under ultrasound monitoring, as shown in Fig. 1A. The delivery of the E2 waveform resulted in the formation of electrolytic products, which are seen in ultrasound, as the bright regions around the electrodes in Fig. 1B. The generator described in the materials and method section performed as designed. A detailed description on the performance of the generator can be found in the materials and methods section.

Two typical E2 waveforms delivered by the E2 device are shown as dashed lines in Fig. 1C. In this study, the electric potential between electrodes was kept below 1,000 V to

Table 1  Statistical summary of histologically observed effects on cells and tissue after applying different E2 parameters. The "Parameters" column group specify how the E2 energy was applied (charge, field, number of waves). The "Sample Info" column group specifies the sample location ("close" refers to samples near the electrodes, whereas "middle" refers to samples taken from the midline between the electrodes). The "Slice Ref" column refers to the image names as stored in https://doi.org/10.6084/m9.figshare.8061827.v1. The "Histologically observed effects" column lists effects that are of interest for deduction of ablation completeness (1st to 5th column counted from left to right) and for tissue selectivity (6th and 7th column). In the notation "X of Y", X denotes the number of evaluated pathology images where the specified effect was observed (compared to control) in a majority of the tissue visible on the image. Y denotes the number of images where the effect could have been observed. Time gaps between E2 treatment and pathological fixation (24 h and 72 h after) are separated by a dash where appropriate.

| Parameters | | | Sample Info | | Histologically observed effects | | | | | | | |
|---|---|---|---|---|---|---|---|---|---|---|---|---|
| Charge per wave | Voltage-to-Distance ratio | Count of waves applied | Sample location | Slice name reference 24 h/72 h | Condens. of hepatocyte cytoplasm after 24 h (compared to control) | Condensed or pyknotic nuclei after 24 h (compared to control) | Hemorrhage or edema after 24 h (compared to control) | Damaged sinusoids after 24 h (compared to control) | Cellular structure disruption 24 h/72 h (compared to control) | At least one intact bloodvessel after 24 h/72 h | At least one intact bile duct after 24 h/72 h | Increase in fibrous tissue after 72 h (compared to control) |
| 35 µF | 1,000 V/cm | 1 | Close | p1l1a / NA | 6 of 6 | 5 of 5 | 6 of 6 | 6 of 6 | 6 of 6/NA | 5 of 5/NA | 1 of 1/NA | NA |
| | | | Middle | p1l1 a,b,c (T,I) / NA | 20 of 20 | 20 of 20 | 20 of 20 | 20 of 20 | 20 of 20/NA | 20 of 20/NA | 20 of 20/ NA | NA |
| 90 µF | 1,000 V/cm | 1 | Close | p1l2 / p2l1 | 6 of 6 | 5 of 5 | 6 of 6 | 6 of 6 | 6 of 6/4 of 6 | 6 of 6/6 of 6 | 3 of 3/6 of 6 | 6 of 6/4 of 6 |
| | | | Middle | p1l2 / p2l1 | 6 of 6 | 5 of 5 | 6 of 6 | 6 of 6 | 6 of 6/0 of 6 | 6 of 6/6 of 6 | 6 of 6/6 of 6 | 6 of 6/0 of 6 |
| 122 µF | 1,000 V/cm | 1 | Close | p1l6b / p2l2a | 6 of 6 | 5 of 5 | 6 of 6 | 6 of 6 | 6 of 6/18 of 18 | 6 of 6/18 of 18 | 6 of 6/12 of 12 | 18 of 18 |
| | | | Middle | p1l6b / p2l2a | 6 of 6 | 5 of 5 | 6 of 6 | 6 of 6 | 6 of 6/5 of 5 | 5 of 5/5 of 5 | 4 of 4/5 of 5 | 6 of 6 |
| 122 µF | 1,000 V/cm | 3 | Close | p1l5b / NA | 6 of 6 | 5 of 5 | 6 of 6 | 6 of 6 | 6 of 6/NA | 6 of 6/NA | 6 of 6/NA | NA |
| | | | Middle | p1l5b / NA | 6 of 6 | 5 of 5 | 6 of 6 | 6 of 6 | 6 of 6/NA | 6 of 6/NA | 6 of 6/NA | NA |
| 122 µF | 660 V/cm | 1 | Close | NA | NA | NA | NA | NA | NA | NA | NA | NA |
| | | | Middle | p1l8 / NA | 2 of 6 | 2 of 5 | 0 of 6 | 2 of 6 | 6 of 6/NA | 6 of 6/NA | 6 of 6/NA | NA |
| 122 µF | 660 V/cm | 2 | Close | NA | NA | NA | NA | NA | NA | NA | NA | NA |
| | | | Middle | p1l9b / NA | 6 of 6 | 5 of 5 | 6 of 6 | 6 of 6 | 6 of 6/NA | 6 of 6/NA | 6 of 6/NA | NA |
| 122 µF | 660 V/cm | 3 | Close | NA | NA | NA | NA | NA | NA | NA | NA | NA |
| | | | Middle | p1l10a / NA | 6 of 6 | 5 of 5 | 6 of 6 | 6 of 6 | 6 of 6/NA | 6 of 6/NA | 6 of 6/NA | NA |
| 218 µF | 1,000 V/cm | 2 | Close | NA | NA | NA | NA | NA | NA | NA | NA | NA |
| | | | Middle | p2l8 / NA | NA | NA | NA | NA | NA/6 of 6 | NA/6 of 6 | NA/6 of 6 | 6 of 6 |
| 122 µF | 1,000 V/cm | >=1 | Middle | p1l6b, p1l5b | 12 of 12 | 10 of 10 | 12 of 12 | 12 of 12 | 12 of 12 | 11 of 11 | 10 of 10 | NA |
| 122 µF | >=660 V/cm | >=2 | Middle | p1l5b,p1l9b, p1l10a | 18 of 18 | 15 of 15 | 18 of 18 | 18 of 18 | 18 of 18 | 18 of 18 | 18 of 18 | NA |

eliminate possible detrimental effects based on IRE pulses, which can produce electrical discharge across gases produced by electrolysis at the electrodes (*Guenther et al., 2015*). Higher voltage pulses were not part of this in-vivo study. The chosen parameters resulted in a maximal nominal electrical field strength of 1,000 V/cm. This voltage is typical to voltages used for tissue ablation with reversible electroporation and chemotherapeutic drugs such as bleomycin (ECT) (*Mir et al., 2006*). It was found that this voltage obviates the need for pharmacological muscle relaxation, affecting only a single, manageable muscle contraction in the 80 kg pigs. Figure 1C shows two 300 millisecond long waveforms, recorded when delivered by the device. The waveform delivers first 1,000 V, after which it decays rapidly. As mentioned earlier, the rapid decay was designed to avoid electrical discharge across the electrolytic products, a phenomenon typical to IRE pulses (*Guenther et al., 2015*). This is followed by a relatively shallow decay to facilitate the delivery of the charge generated electrolytic products. The designed waveform resembles an exponential decay pulse. The solid red line in Fig. 1C shows the shape of a single exponential decay pulse that would have delivered the same charge as the E2 waveform in the figure (best fit).

## The minimum electrical charge per volume required for complete tissue ablation between the electrodes (bridged ablation)

An important parameter in designing the E2 waveform is the amount of charge delivered. In our design, the charge is delivered from the discharge of a capacitor. The amount of charge in the capacitor is given by the capacity of the capacitor and the voltage. The initial voltage is set to 1,000 V. The extent of tissue ablation was evaluated for three capacitors of: 35 µF, 90 µF, 122 µF. Experiments were done with two parallel electrodes, separated by 1 cm and with an exposure length of 1.5 cm. The results are shown in Fig. 4. The different panels display the appearance of the gross histology (Figs. 4F, 4N, 4V), US images (Figs. 4H, 4P, 4X) and H&E and Mas stained magnified images (Figs. 4B–4E, 4J–4M, 4R–4U) from a central location between the electrodes (red square in gross pathology) and an area that served as control (blue square in gross pathology). The analysis is focused on the midsection, which is the area with both the smallest field strength (electroporation) and least amount of produced electrolytic species. Fig. 4A shows the results of the application of 35 µF, row B 90 µF and row C 122 µF. The gross pathologies of Figs. 4N and 4V with higher capacity show a dark red area between the electrodes (black arrows in Figs. 4N and 4V), indicating the region in which the tissue was affected (called bridged region), which is due to accumulation of red blood cells. A demarked area in the midsection is also visible in the application with 35 µF (black arrow in Fig. 4F), though less severe and without macroscopically visible hemorrhage. However, microscopic examination of the cells within the midsection (Figs. 4B–4E, 4J–4M, 4R–4U) shows comparable cellular disruption at all three capacities, with severe signs of cellular ablation injury with condensation of hepatocyte cytoplasm, condensed nuclei and hemorrhage throughout all six micrographs. In comparison, control images (Figs. 4B, 4C, 4J, 4K, 4R, 4S) at same magnification show intact sinusoids, no coagulated blood and regular-sized nuclei. The bright area visible on ultrasound (Figs. 4H, 4P, 4X), which were taken during the experiment, was caused by the products of electrolysis generated near the electrodes.

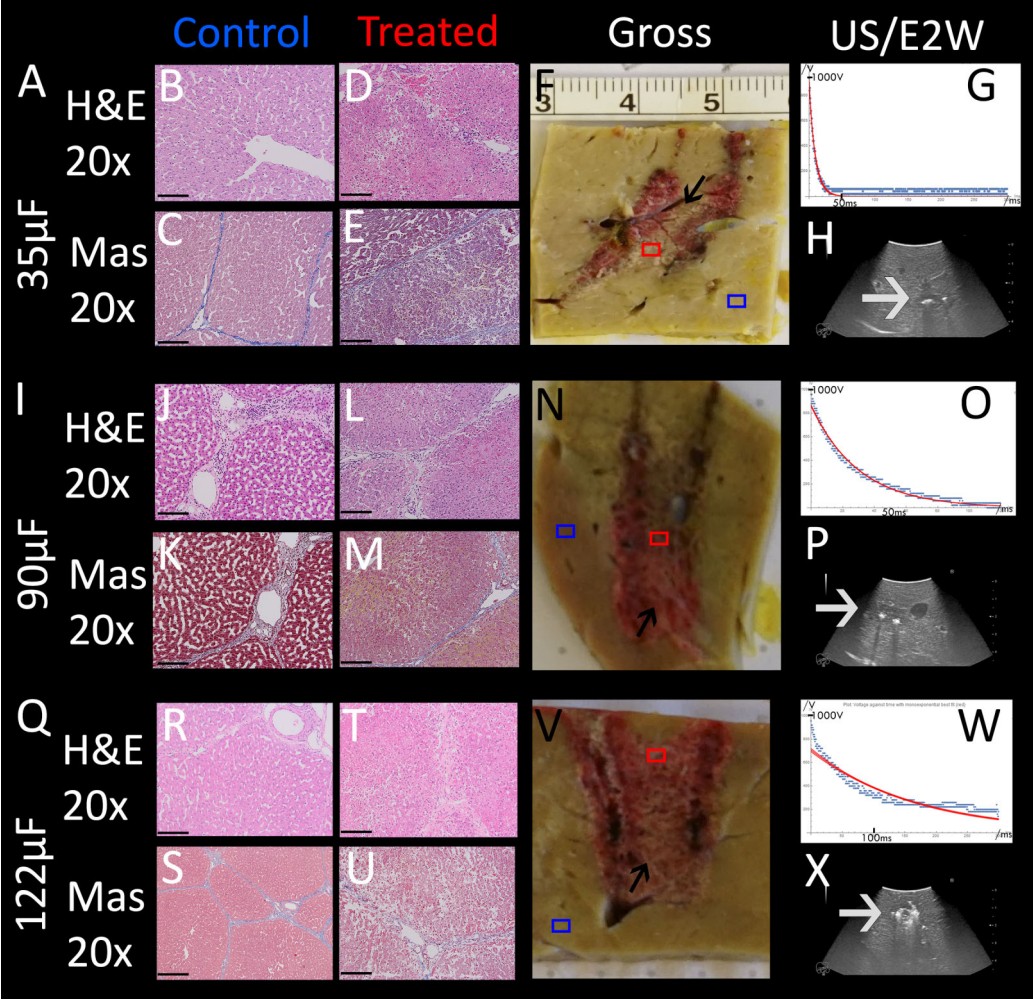

**Figure 4 Effects of the same application of an E2 waveform (E2W) (1 cm distance, 1.5 cm exposure, 1,000 V) at different applied charges.** The histological slides (H&E B, D, J, L, R, T and Masson's tri-chromatic stain C, E, K, M, S, U) show 20× magnification of the mid-section area between the probes (red square) and control (blue square), location is shown in gross pathology in the "Gross" column F, N, V. Ultrasound images as well as the trace of voltage as a function of time of the applied E2W during the experiment are shown in H, P, X. (A) 35 μF. (I) 90 μF and (Q)122 μF. Scale bars indicate 100 μm. Microscopically, ablation in the mid-section area was confirmed for all charges with increasing degree of hemorrhage.

These areas are visible in all experiments (white arrows in Figs. 4H, 4P, 4X). Images on top of the US snapshots in the last column show the E2 waveform that was applied. While the E2 waveforms shown in Figs. 4O and 4W look similar, the applied charge in A6 was delivered in a shorter amount of time.

## Determination of the minimum number of E2 waveforms required for complete tissue ablation between the electrodes (bridged ablation)

The purpose of this part of the study was to evaluate the effect of E2 waveform repeats on the extent of the ablation zone and in particular, to asses if one single E2 waveform is sufficient to ablate all the cells within the volume between the electrodes. The E2 waveform

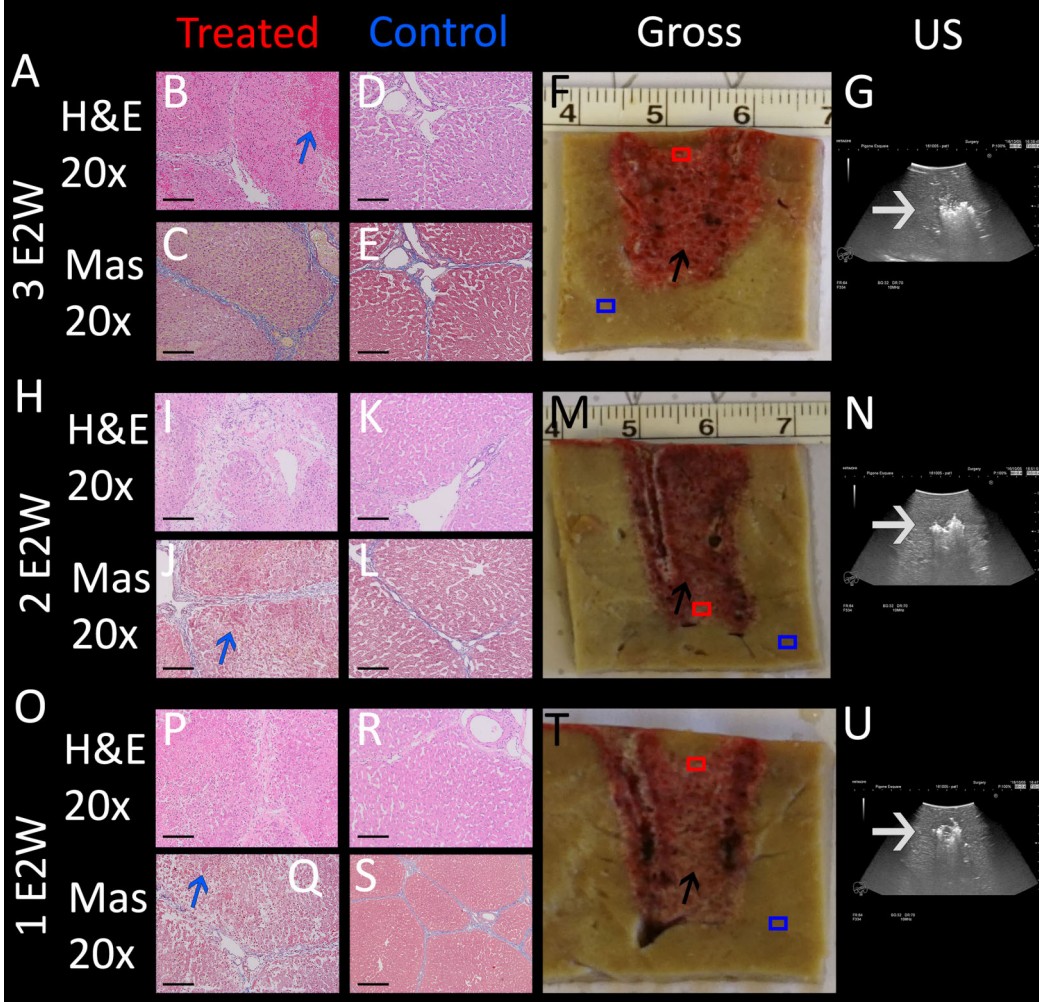

**Figure 5  E2 application at 1 cm distance and 1.5 cm exposed length. Treatment parameters: 1,000 V, 122 μF.** Effects of (A) three E2 waveforms (E2W), (H) two E2 waveforms and (O) one E2 waveform. The histological slides (H&E B, D, I, K, P, R and Masson's trichromatic stain C, E, J, L, Q, S) show 20× magnification of the area between the probes (red square) and control (blue square), as shown in gross pathology in the "Gross" column F, M, T. US snapshots during treatment are shown in G, N, U. Scale bars indicate 100 μm.                           

is shown in Fig. 1C. Multiple ablation zones were created in different regions of the liver of an 80 kg pig. The pig was sacrificed after 24 h and the data evaluated as described in the method section.

The results are shown in Fig. 5, comparing between a protocol that delivered one E2 waveform, two E2 waveforms separated by 30 s and three E2 waveforms separated by 30 s. The waveforms were delivered between two electrodes with a distance of 1 cm and an exposure length of 1.5 cm. The charge was delivered from a source of 122 μF and 1,000 V. Gross pathology (Figs. 5F, 5M, 5T) clearly shows a demarked area in the midsection of all applications, which visualizes a bridged treated region in tissue (black arrows). Again, the dark appearance is due to accumulation of red blood cells. Microscopic evaluation was by means of H&E and Mas, with the location of the control and central
treatment sites shown as blue and red squares in the macroscopic images, respectively. The midsection of the ablation zone (Figs. 5B, 5C, 5I, 5J, 5P, 5Q) in all three experiments reveals total loss of cellular infrastructure, which is visible throughout all microscopic images of the treated area, indicating that the treatment effect on a cellular level correlates strongly with the demarked area visible in gross pathology. Also visible throughout all microscopic images are the affected shrunken nuclei. Examples are shown in blue circles in both control and treated areas for direct comparison in the microscopic images of higher resolution which can be found in the https://doi.org/10.6084/m9.figshare.8061827.v1. Heavy hemorrhage is visible in all treated liver parts (e.g., are pointed at with blue arrows) however, the severity increases with increased number of applied waveforms, with the strongest effect visible in Fig. 5B (blue arrow). The electrolytic products which are generated during treatment can be seen as bright areas on ultrasound (white arrows in Figs. 5G, 5N, 5U). This series of experiments shows that all three applications have resulted in successful ablation of the tissue between the probes. Notable is that one single E2 waveform was sufficient, and had comparable results to the application with more waveforms.

## Determination of the effect of the electric field magnitude on the ablation protocol

In most of our experiments a voltage over distance ration of 1,000 V/cm was chosen because this ration is commonly used for ECT. A 3 kV system like the one tested here would therefore be capable to produce an ablation field with a diameter of 3 cm, allowing for relatively large treatment fields. However, in this section it is evaluated if lower fields can to some degree be compensated with more electrolysis and/or more waveforms. Therefore, experiments were carried out with electrodes spaced 1.5 cm with 1.5 cm exposure length and the 122 μF capacitors charged to 1,000 V, yielding a voltage to distance ratio of 670 V/cm. In three separate experiments one, two and three waveforms were delivered, respectively. The pig was sacrificed after 24 h and the data evaluated as described in the method section. The results are summarized in Fig. 6, with macroscopic images in the top row, and H&E (B, E, H) and Mas staining (C, F, I) of the central part of the created lesion at 20× magnification. Macroscopic images show that the application of one E2 waveform at 670 V/cm results in a non-bridged region (red square in Fig. 6A), while two E2 waveforms created a demarked area in the midsection (red square in Fig. 6D). The most severe effect is visible at three E2 waveforms, where a clear bridged ablation zone can be seen (red square in Fig. 6G). This gradual effect of ablation can be confirmed in the microscopic assessment: In both H&E and Mas stained images which show the results of one applied E2 waveform, widened sinusoids are visible with intact cells, and nuclei appear unaffected. With increasing numbers of applied waveforms, the damage to the tissue becomes more severe. The microscopic images in the middle column show condensed nuclei and disruption of cellular structure (both visible throughout the entire microscopic panels). The application of three E2 waveforms additionally results in complete ablation injury with more severe hemorrhage (G–I), but a sufficient cellular treatment effect is comparable at both two and three waveforms applied. As expected, a closer look at the macroscopic images reveals that the "burned" radius

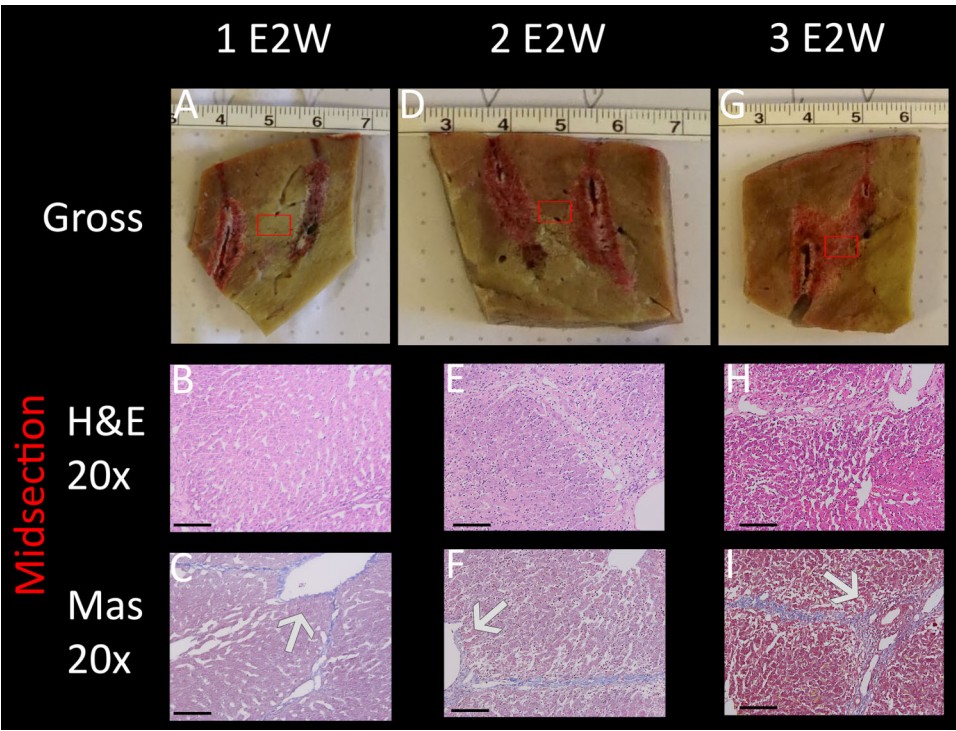

**Figure 6 Assessment of compensation for lower voltage-distance-ratio (670 V/cm) with more waveforms and more electrolysis.** 1,000 V was used for 1.5 cm distance (instead of 1.0 cm) and 1.5 cm exposure length. 1 E2W (A–C): Not bridged. 2 E2W (D–F): macroscopically not bridged but microscopically all cells appear to be dead in the mid-section after 24 h. 3 E2W (G–I): Bridged ablation zone with identical pathology to 1,000 V at 1 cm. Arrows show intact vessels. Scale bars indicate 100 µm.                               

around the electrodes slightly increased in size with increased numbers of waveforms applied. Hence, a 670 V/cm voltage-distance ratio is sufficient for ablation when enough electrolytic species are distributed by using several waveforms. The ablation zone roughly translates to 250 V/cm field as opposed to IRE, where >700 V/cm appears to be the minimum (depending on tissue, number of pulses etc.) It should also be noted that even though complete tissue ablation was successful in the last two applications, the structure of the vessels remained intact (arrows in Figs. 6C, 6F and 6I).

## The effect of E2 waveforms on important tissue structure

Vessels and nerves are among the vital structures which are often within the treatment field and should be spared if possible. Intact vessels with undisrupted endothelial layer are a necessity for the recovery of liver tissue and can thus accelerate the healing process after treatment. To examine the effect of E2 waveforms on these tissue structures, several experimental sites which included arteries, bile ducts and nerves were assessed. Fig. 7 summarizes the results from experiments that were carried out with the application of a single E2 waveform with the following parameters: 1 cm distance, 1.5 cm exposure length, 1,000 V and 122 µF. The focus was on structures in close approximation to an electrode due to higher potential of damage to the vessels. The top row shows H&E stained

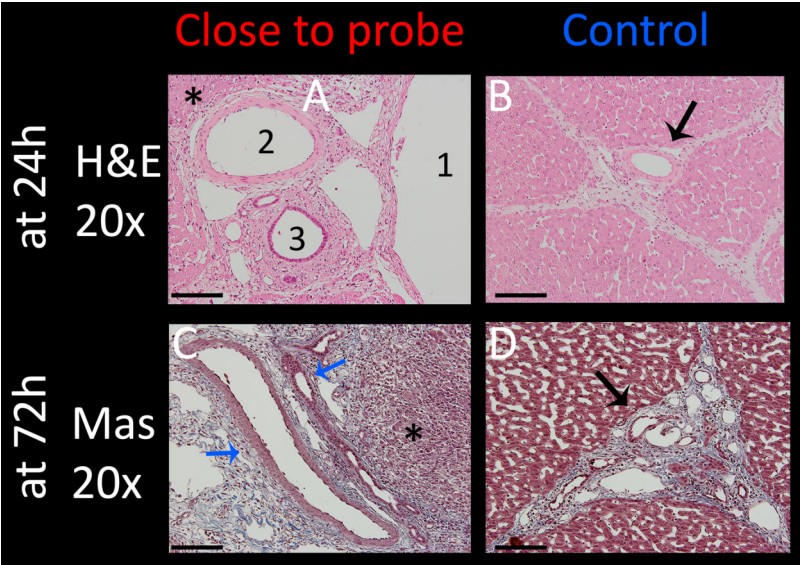

**Figure 7 Effect of a single E2 waveform on vessels.** Experiment parameters were: 1,000 V, distance of 1 cm, exposed length of 1.5 cm, 122 µF capacity. Left column (A and C): 20× magnification of the treatment site close to an electrode, right column (B and D): control (arrows point at control vessel). Top row (A and B) shows H&E stained images after 24 h with 1 = portal vein, 2 = artery, 3 = bile duct, all of which are intact. Bottom row (C and D) shows pathological slides stained with Masson trichrome, illustrating intact vein and bile duct (blue arrows). Scale bars indicate 100 µm.

images of 20× magnification of tissue which was harvested after 24 h. The artery (Fig. 7A2) which was in the center of the treated field presents with morphologically intact endothelial layer and remaining vascular smooth muscle cells. All other vessels (Fig. 7A, 1 = portal vein, 3 = bile duct) show preserved infrastructure without structural damage, while the surrounding tissue shows clear signs of ablation effects, with signs of edema, hemorrhage and condensed nuclei (asterisks in Fig. 7A). Figure 7B shows the control, which includes a control vessel (arrow) that can be seen as comparable to that of the treated tissue. The Figs. 7C and 7D show Mas stained slides of tissue which was harvested after 72 h. Again, clear ablation signs are visible in the surrounding tissue, with complete loss of structural integrity, shrunken nuclei throughout the displayed micrograph and heavy hemorrhage (asterisks). Vessels, however, showed intact elastic fibers and preserved vessel wall (blue arrows). These results imply that the procedure is capable of sparing vessels.

## Understanding additional mechanisms of damage superimposed on the damage due to the combination electroporation and electrolysis

Electric fields can cause a variety of effects simultaneously. For example, some thermal effects caused by Joule heating occurs simultaneously with IRE and E2. It is to be anticipated that these additional effects will be most pronounced near the electrodes. As mentioned in the introduction, thermal effects are highly undesirable, because they ablate every tissue component in the treated area. It is thus important to know the extent of these additional effects that are superimposed on the combination electroporation and

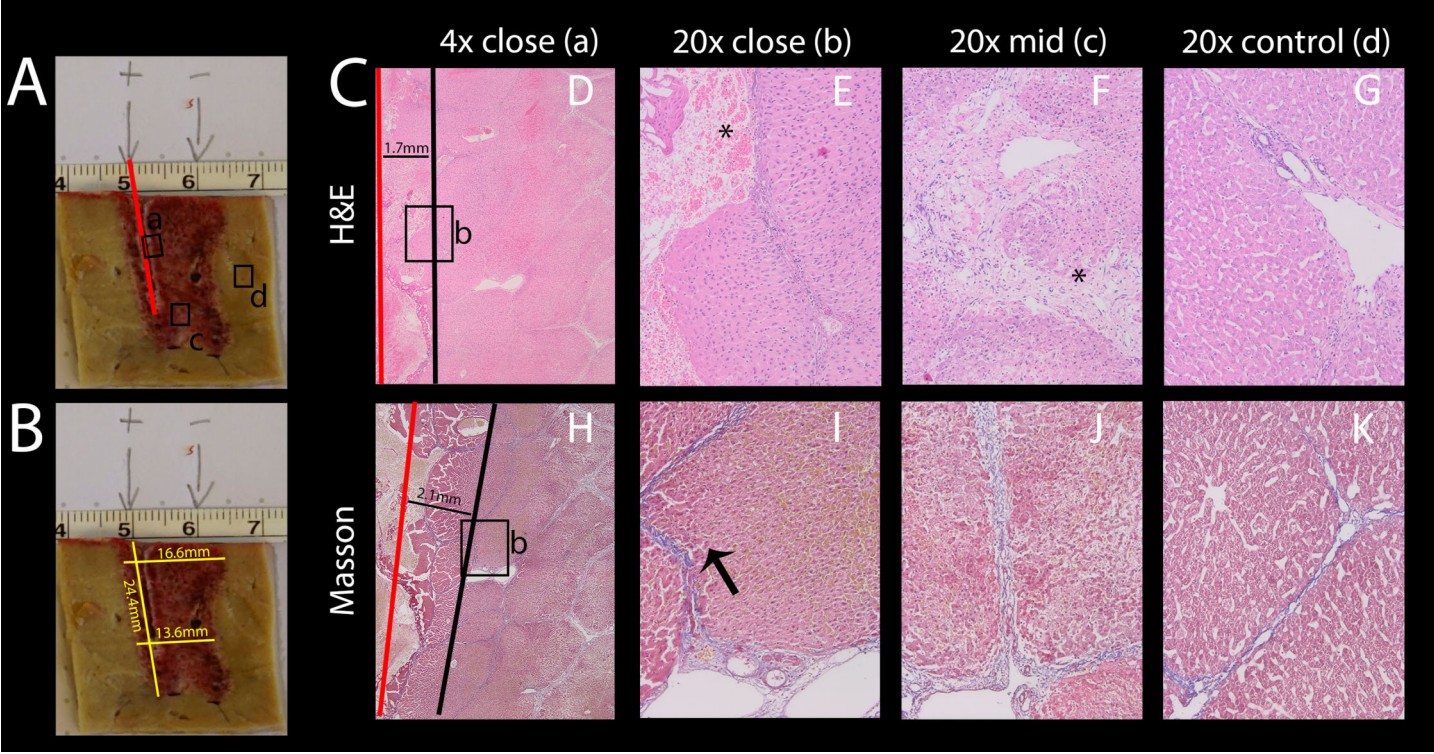

**Figure 8 Effects of 2 E2 waveforms at 1 cm distance, 1.5 cm exposure length, 1,000 V and 122 µF at different magnifications and distances from the probe.** (A) Macroscopic image of the resulting lesion, showing the location of the close-ups in C. (B) Lesion dimensions. (C) Micrographs showing the damage near the electrode entrance (red line) at 4× (D, H) and 20× magnification (E, I) compared to the midsection (F, J) and compared to control (G, K); H&E staining (D–G) and Masson's trichromatic staining (H–K).

electrolysis, as it is of high relevance for treatment planning. While these effects are inescapable, proper design can minimize the severity. It was therefore our goal to find the approximate radius in which tissue infrastructure is clearly damaged due to these effects. To study this, an example with excessive electrolysis and pulsing was chosen. The lesion was created using 2 x 122 µF E2 waveforms at 1,000 V. The distance between the electrodes was 1 cm and the electrode exposure was 1.5 cm of length. As in all experiments, the probes were oriented parallel to each other. The histological results after 24 h are shown in Fig. 8. The macroscopic image in Fig. 8A shows a bridged ablation zone, with dimensions shown in Fig. 8B, indicating a homogeneous severity of the ablation effect. The location of the microscopic images in Fig. 8C are marked in Fig. 8A. Microscopic images at close approximation to the electrode are shown at 4× (Fig. 8D) and 20× magnification (Fig. 8E). A micrograph of the midsection of the ablation zone is shown in Fig. 8F, while Fig. 8G shows control slides. Pathological slides stained with H&E (Figs. 8D–8G) show tissue integrity damage by slight cracks and fissures (marked with asterisks) as well as deformed and shrunken nuclei (Figs. 8E, 8F). Mas staining (Figs. 8H–8K) however additionally reveals severe acute hepatocellular necrosis with coagulated blood at 2 mm proximity to the electrode that can be observed when denaturation due to heat takes place, with complete loss of tissue integrity.

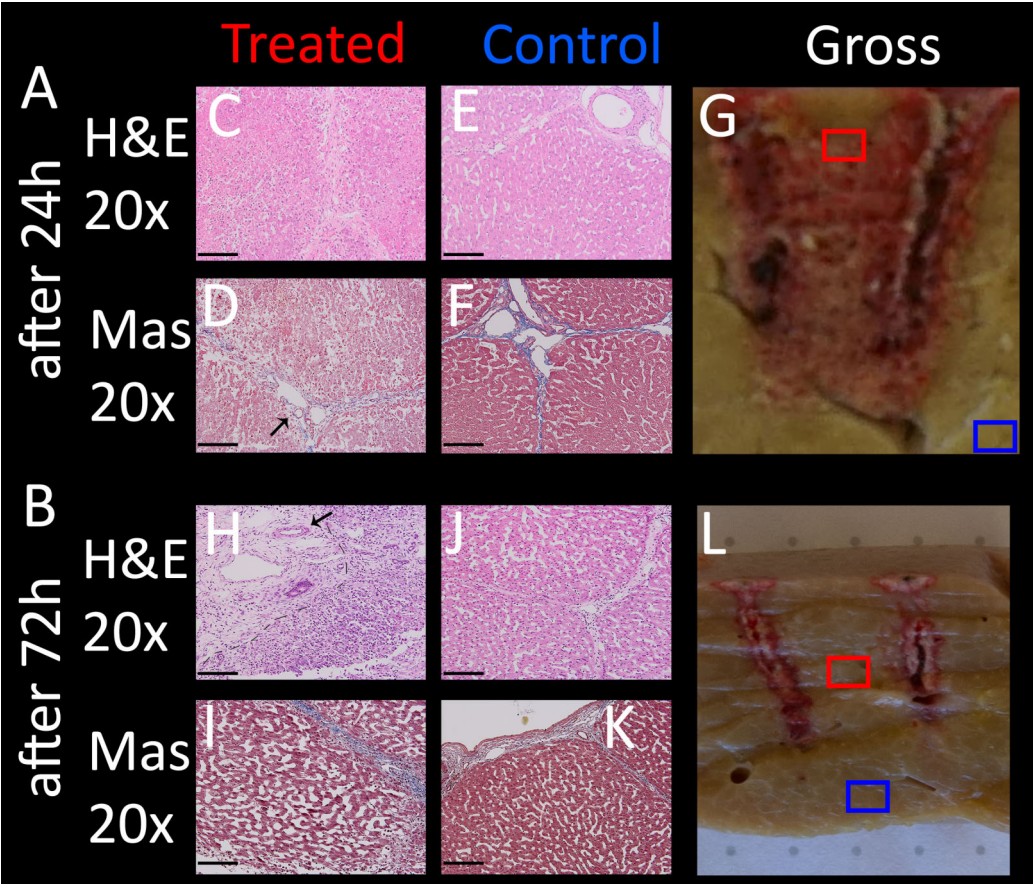

**Figure 9 Comparison of the application of a single E2W with 1 cm spacing between probes, 1.5 cm exposure length, 1,000 V and 122 μF, at different recovery times (24 h, 72 h).** On the left are shown histological images which were taken from the midsection (C, D, H, I), the middle column (E, F, J, K) shows control images (both at 20× magnification), while the images on the right (G, L) show macroscopic images, the boxes indicating where the lesion was analyzed microscopically. H&E (C, E, H, J) and Masson trichrome staining (D, F, I, K) were used. (A) After 24 h (B)After 72 h. Sample position boxes in the gross images are approximations. Scale bars indicate 100 μm. The regenerative zones are composed of non-homogenous cell populations, elongated and spindle shapes cells and ductular cells (arrow).

This experiment shows that an area of approximately 2 mm radius from the electrode gets affected by thermal and/or electrolytic damage.

## Temporal histology after treatment with an E2 waveform

To compare treated liver tissue at different recovery times, a lesion created with identical parameters was analyzed in pigs sacrificed after 24 and 72 h. The pathological results are illustrated in Fig. 9, with the right column showing gross pathology with indication of the microscopic samples, the left column showing micrographs of the treated area at 20× magnification, and the middle column showing control samples of the same slice outside the treated area at identical magnification. Treated tissue after 24 h (Fig. 9A) reveals dilated sinusoidal spaces due to either edema and/or hepatocellular swelling throughout the micrographs on the left. The nuclei are contracted and hemorrhage is

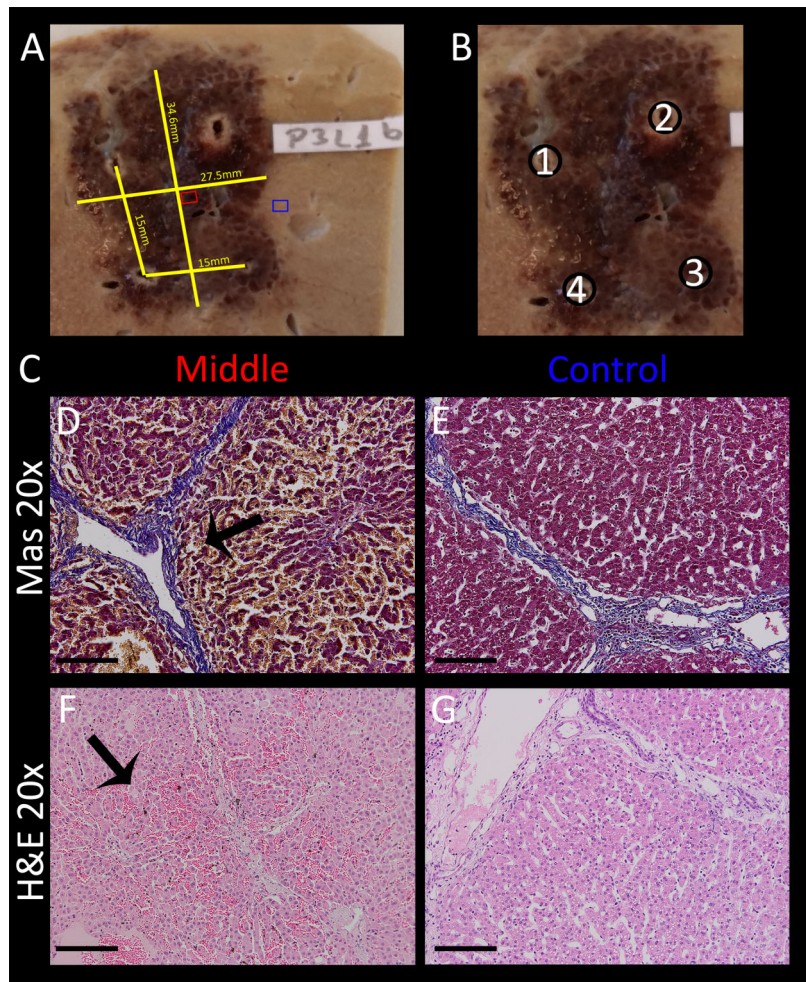

**Figure 10 Volumetric ablation with four electrodes spaced with 1.5 cm distance and 1.5 cm exposure length.** $C$ = 218 μF and $U$ = 1,000 V. The total delivery time of the E2W was less than 10 s including capacitor charging time. Ablation was performed without muscle relaxation. (A) Gross pathology, showing lesion dimensions (yellow), location of midsection slide (red box) and control slide (blue box). (B) Schematic with the associated electrode numbers. The probe pair programing was set to 1–2, 3–2, 3–4, 1–4, 3–1 and 2–4, with first # = anode and second # = cathode. (C) Masson's trichrome staining (D, E) and H&E staining (F, G) of treatment area (D, F) and control area (E, G) at 20× magnification. Micrographs of treated lesion show severe acute hepatocellular necrosis with coagulated blood in the sinusoids. Scale bars indicate 100 μm.

visible in the entire area compared to control (E, F, J, K). The severity of cellular ablation is high, as the cellular matrix is completely disrupted. In contrast, micrographs of lesions 72 h post treatment (Figs. 9H–9K) do not seem to show condensed nuclei compared to control images. Cellular structures appear to be disrupted (e.g., see cells on the right side of the dashed line in Fig. 9H, but not as severe compared to 24 h post treatment, though direct comparison is difficult due to different grades of dilatation of the sinusoids of both controls and treated lesions. Note how vessels remain intact in these experiments as well (arrows), while there appears to be an increase in fibrous tissue in the top left corner of the micrograph (left side of dashed lines, see high resolution images

in https://doi.org/10.6084/m9.figshare.8061827.v1), indicating regeneration processes of the liver, which is a known phenomenon also from other electroporation based ablation technologies (*Chang, Zhou & Rubinsky, 2017*; *Golberg et al., 2016*).

### Simulation of a typical clinical procedure

In typical clinical procedures with IRE and ECT, the tumor is surrounded by electrodes and a series of electroporation pulses are delivered between the pulses. The following experiment was designed to simulate such a procedure with E2, while it also evaluated the generator's ability and stability to provide E2 waveforms for the creation of a large lesion without muscle relaxant and in less than 10 s (thus non-ECG triggered). The electrodes were placed in a square configuration parallel to each other with a separation of 1.5 cm and an exposure length of 1.5 cm. A high capacitance of 218 µF and 1,000 V was used (=670 V/cm Voltage distance ratio). The probe pair programing was set to 1–2, 3–2, 3–4, 1–4, 3–1, 2–4, with the first number being the anode and the second the cathode. The capacitance was fully discharged with an E2 waveform over a period of roughly 300 ms each. The approximate volume of ablated tissue was 25 mm x 32 mm x 25 mm = 20 cm$^3$. This is only about 1/6th of the volume the device is designed instantly ablate with four electrodes, but high energy and maximum and lesion size tests were not part of this study design. The total time for charging and E2 delivery was approximately 10 s. The pig was sacrificed after 2 h. One slice of the resulting lesion including the dimensions is shown in Fig. 10A, the electrode configuration in Fig. 10B. Microscopic images (Fig. 10C) of control (Figs. 10E, 10G) and treated area (Figs. 10D, 10F) are shown at 20× magnification both in H&E and Mas staining. Even though direct comparison of treated and untreated tissue does not reveal pyknotic nuclei after 2 h (Figs. 10F vs. 10G, which is a comparison of treated and untreated liver tissue, show similar sized nuclei), swelling of the cells, hepatic necrosis with hypereosinophilic cytoplasm with resulting dilated sinusoids is visible in the left column, which is best visible in the center of micrograph Fig. 10F in (arrow). Note how the vessels are completely intact (Fig. 10D, arrow).

## DISCUSSION AND CONCLUSION

A statistical summary of the histologically evaluated samples for different E2 parameters can be found in Table 1. The table is sorted by the parameters used (capacity in µF, the Voltage-to-Distance ratio, the number of waves). The sample location refers to the area where the histology was evaluated relative to the position of the two electrodes: "Close" means 1–3 mm from either electrode where the electric field as well as the electrolytic and heat effects are expected to be high; "Middle" refers to the mid line between the electrodes where the electrical field as well as electrolytic product density and heating is theoretically minimal. The "histologically evaluated effects" column lists the significant changes in cell/tissue structure. For the last four, in cases where 72 h data was available, the information listed was separated with a slash. For each effect, the number of images where these effects were visible are listed in conjunction with the total number of images where these effects could have been observed. Thus, for example, "6 of 6" means that in six available images, this specific effect has been observed in all six of them (=100%).

All "effects" are counted if the described property is clearly observable relative to the control images of each sample and present on most of the tissue visible on the image. In cases of intact vessels and bile ducts, at least one occurrence was required to be counted as a positive. The column "slice ref" lists the slice file names as can be found in the https://doi.org/10.6084/m9.figshare.8061827.v1.

It is apparent that all tissue changes are observable very uniformly across all parameters. The lowest two lines of Table 1 summarize the clinically relevant quintessence: For all cases where 122 μF was used in conjunction with 1,000 V/cm Voltage-to-Distance ratio, the ablation was complete and successful on all evaluated pathology images. Additionally, all ablations which employed two or more discharge waves and ≥660 V/cm Voltage-to-Distance ratio, were also complete according to all evaluated images. Finally, at least partial preservation of vessels and bile ducts were observed in all cases. The confirmation of these parameters, which were designed to deliver the right amount of electrolysis and electroporation for E2 ablation of clinically practical geometry, was a main purpose of this study.

The combination of in vivo produced electrolytic species and reversible electroporation—E2—allows large volume tissue ablation with a single E2 waveform. This experiment was designed to determine boundaries required for further optimization of the waveform for maximum safety and clinical utility. The clinically relevant result for different parameters are summarized in Table 1. The requirement for successful ablation is high enough voltage for inducing reversible electroporation and producing sufficient electrolytic species by choosing an appropriate delivered charge per volume. The ablation appears to primarily affect cells, keeping the tissue infrastructure intact, a property which is similar to non-thermal IRE. It can therefore be assumed that the E2 treatment is likely to have a similarly advantageous toxicity profile for tissue ablation. This assumption is supported by the fact that vessel structures within the treatment area remained intact, which is a critical factor for tissue recovery. The radius of tissue degeneration near the electrodes due to heat is relatively small, thus decreasing factors that may lead to uncontrollable tissue damage. Additionally, an increased number of delivered E2 waveforms does not seem to cause more tissue infrastructure damage. This makes treatments which require several electrodes to produce a specific ablation geometry feasible. Effective ablation with 670 V/cm voltage to distance ratio with needle type electrodes was shown. With this property, designing the waveform in terms of charge and voltage is intrinsically safer and does not limit exposure length. Whilst high-energy tests were not part of this study, the results confirm that the potential ablation volume with the voltage and capacitance specification of the generator used would be well above 100 cm$^3$ in under 10 s with four electrodes. Muscle relaxation in pigs was not required, because the waveform is mostly delivered before muscles can contract; the single contraction was manageable without additional medication, again reducing a potential source of additional toxicity of the treatment. The technology appears reliable, as several different protocols lead to similar lesions. This makes E2 an effective tool for minimally invasive IGA in liver, and is particularly advantageous in cases where size of the lesion and speed of the procedure matters, and when preservation of vessels is of crucial importance.

### Funding
The authors received no funding for this work.

### Competing Interests
Boris Rubinsky is an Academic Editor for PeerJ. Enric Guenther, Nina Klein, Paul Mikus, Michael Stehling, Boris Rubinsky are employees of InterScience GmbH, Lucerne, Switzerland (a company in the field of tissue ablation by electroporation). Franco Lugnani and Matteo Macchioro are employed by Hippocrates D.O.O. (a medical office/company). All other authors have no competing interests.

### Author Contributions
- Enric Guenther conceived and designed the experiments, performed the experiments, analyzed the data, contributed reagents/materials/analysis tools, prepared figures and/or tables, authored or reviewed drafts of the paper, approved the final draft.
- Nina Klein conceived and designed the experiments, performed the experiments, analyzed the data, contributed reagents/materials/analysis tools, prepared figures and/or tables, authored or reviewed drafts of the paper, approved the final draft.
- Paul Mikus conceived and designed the experiments, contributed reagents/materials/analysis tools, authored or reviewed drafts of the paper, approved the final draft.
- Florin Botea performed the experiments, authored or reviewed drafts of the paper, approved the final draft.
- Mihail Pautov performed the experiments, prepared figures and/or tables, approved the final draft.
- Franco Lugnani performed the experiments, contributed reagents/materials/analysis tools, authored or reviewed drafts of the paper, approved the final draft.
- Matteo Macchioro performed the experiments, contributed reagents/materials/analysis tools, authored or reviewed drafts of the paper, approved the final draft.
- Irinel Popescu contributed reagents/materials/analysis tools, authored or reviewed drafts of the paper, approved the final draft, preparation and infrastructure of the experimental facilities.
- Michael K. Stehling conceived and designed the experiments, contributed reagents/materials/analysis tools, authored or reviewed drafts of the paper, approved the final draft.
- Boris Rubinsky conceived and designed the experiments, analyzed the data, contributed reagents/materials/analysis tools, authored or reviewed drafts of the paper, approved the final draft.

### Animal Ethics
The following information was supplied relating to ethical approvals (i.e., approving body and any reference numbers):

The Ethics Committee of Fundeni Clinical Institute and the Bucharest Sanitary-Veterinary Authority (no. 316) approved this study.

## Patent Disclosures

The following patent dependencies were disclosed by the authors:

P249767.WO.01: PCT/US2014/065794 filed Nov. 14, 2014

P249766.WO.01: PCT/US2014/065783 filed Nov. 14, 2014.

## Data Availability

The pathology images are available at Figshare:

Guenther, E (2019): supplements.zip. figshare. Journal contribution.
DOI 10.6084/m9.figshare.8061827.v1.

Additional ultrasound and oscilloscope data and photographs are available at Figshare: Guenther, E (2019): Supporting information - US, Osci, Fotos. figshare. Journal contribution. DOI 10.6084/m9.figshare.8100503.v1.

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
