# Peer review of "Toward a clinical real time tissue ablation technology: combining electroporation and electrolysis (E2)"

_PeerJ, doi:10.7717/peerj.7985_

## Round 0.1 · original submission · Minor Revisions

Based on the opinions of two expert reviewers and on my observations, I think that the paper is well written and an interesting work. I think it is an original and important study and suitable for publication in PeerJ after minor revisions.

Reviewer 1 ·

Basic reporting

The paper is well written, refers to relevant literature.
Several figures are used, however a sumamry table reporting different experimental conditions would help the reader.

Experimental design

Experimental conditions are properly described.

Validity of the findings

The methodology is properly decribed and the results interesting.
Beside figures, the authors should make an effort to summarize experimental conditions and results in a table. It might help to evidence the validity of the new technology to quantify the results (i.e. by mesurement of the ablated area) in the different experimental conditions.
The authors might consider to evidence cell induced apoptosis by specific stainign of the samples.

Reviewer 2 ·

Basic reporting

The manuscript is very well written and the length is adequate. It includes a sufficient background and appropriate literature references.
The structure of the article is conform to the format of standard sections. Figures are relevant but they should be graphically improved (see Comments for the authors). The article is self-contained with relevant results to hypotheses.

Experimental design

The content of the study is original and well designed and should make a nice addition to the literature. Methods are described in detail.

Validity of the findings

Impact and novelty are achieved and the conclusions are well stated, linked to original reserach and translational research questions, however the data should be supported by statistical analysis.

Additional comments

Gunther and colleagues presented an interesting work describing a successful and clinically relevant real time tissue ablation technology based on the combination of electroporation and electrolysis. In general, this work can represent an advance in the knowledge about the treatment of solid malignancies based on the use of irreversible electroporation (IRE) and reversible electroporation with chemotoxic drugs, called electrochemotherapy (ECT).

The manuscript is very well written and theorically well designed, however I have the following major concerns regarding this work.


Point 1.
Qualitative data are not associated to any statistical test so the results are not statistically sound and controlled. Please, try to add a statistical analysis of the data emerging from the histological study.

Point 2.
Lines 142-143: The authors cite 19 lesions but it is not provided any description of them. Please, give a short definition of the type of lesions identified in the three pig livers.

Point 3.
Figure 1 and Figure 2 should be inverted in the text following the numerical order.

Point 4.
Overall, the figure need to be graphically implemented. Figure 1, Figure 2C, Figure 3 (third column): please, increase the size of text.

Point 5.
In the legend of Figure 2B a brief description of what the ultrasound images show should be added.

Point 6.
At line 241 the authors tell about “condensation of hepatocyte cytoplasm, condensed nuclei…” but these features are not visible in the histological sections included in the figures at 20x magnification. 40x magnification images can be suggested.

Point 7.
Supplementary files are not specified and visible.

Point 8.
Lines 365-366: the description of Figure 8 does not correspond to the real content of the figure.

In addition, few orthographic errors should be addressed in order to make this work ready for publication. Moreover the English language should be improved to ensure that an international audience can clearly understand your text. Some examples where the language could be improved include:
such as the following ones:
- lines 53-54, 239-240, 243-244, please revise the verb form
- lines 78-79, 350-351, please revise the English form
- lines 85, please consider to use the form “Marshall et al”
- lines 109-110, please revise the English form
- please, use the extended form of the acronyms only the first time they appear in the text, e.g. ECT, Masson’s trhicrome staining (Mas), KCI, etc.

---

## Round 0.2 · accepted · Accept

Your manuscript has now been seen again by our one of our referees, whose comments appear below. In light of their advice, I am delighted to say that we are happy to publish this suitably revised version of your manuscript in PeerJ.

Reviewer 2 ·

Basic reporting

no comment

Experimental design

no comment

Validity of the findings

no comment

Additional comments

The authors have addressed all my concerns.